

# Clinical characteristics and a diagnostic model for high-altitude pulmonary edema in habitual low altitude dwellers

Qiong Li[1,2], Zhichao Xu[1,2], Qianhui Gong[1,2] and Xiaobing Shen[1,2]

[1] School of Public Health, Key Laboratory of Environmental Medicine Engineering, Ministry of Education, School of Public Health, Southeast University, Nanjing, Jiangsu, China
[2] School of Public Health, Department of Epidemiology and Health Statistics, School of Public Health, Southeast University, Nanjing, Jiangsu, China

## ABSTRACT

**Background:** The fatal risk of high-altitude pulmonary edema (HAPE) is attributed to the inaccurate diagnosis and delayed treatment. This study aimed to identify the clinical characteristics and to establish an effective diagnostic nomogram for HAPE in habitual low altitude dwellers.

**Methods:** A total of 1,255 individuals of Han Chinese were included in the study on the Qinghai-Tibet Plateau at altitudes exceeding 3,000 m. LASSO algorithms were utilized to identify significant predictors based on Akaike's information criterion (AIC), and a diagnostic nomogram was developed through multivariable logistic regression analysis. Internal validation was conducted through bootstrap resampling. Model performance was evaluated using ROC curves and the Hosmer-Lemeshow test.

**Results:** The nomogram included eleven predictive factors and demonstrated high discrimination with an AUC of 0.787 (95% CI [0.757–0.817]) and 0.833 (95% CI [0.793–0.874]) in the training and validation cohorts, respectively. Calibration curves were assessed in both the training ($P = 0.793$) and validation datasets ($P = 0.629$). Confusion matrices revealed accuracies of 70.95% and 74.17% for the training and validation groups. Furthermore, decision curve analysis supported the use of the nomogram for patients with HAPE.

**Conclusion:** We propose clinical features and column charts based on hematological parameters and demographic variables, which can be conveniently used for the diagnosis of HAPE. In high-altitude areas with limited emergency environments, a diagnostic model can provide fast and reliable diagnostic support for medical staff, helping them make better treatment decisions.

## INTRODUCTION

Short-term exposure may cause non-acclimatized individuals to suffer from acute mountain sickness, high-altitude pulmonary edema (HAPE), and high-altitude cerebral edema (HACE) following a rapid ascent above 2,500 m (*Sartori et al., 2002*). HAPE represents a life-threatening form of lung injury resulting from exposure to hypobaric

Corresponding author
Xiaobing Shen, xb.shen@seu.edu.cn

hypoxia at high altitudes. Typical symptoms include shortness of breath, fever, tachycardia, hacking cough, cyanosis, and occasionally pink sputum (*Swenson et al., 2002*). HAPE constitutes a form of hydrostatic pulmonary edema wherein the permeability of the alveolar capillaries is altered. There are some possible pathways for the development of early HAPE, one is a traumatic breakdown of the basement membrane of the alveolar capillary barrier caused by high pulmonary arterial pressure, allowing passage of plasma and blood cells, and another pathway may be of transcellular passage through vesicular channels that formed by high intracapillary pressure (*Swenson et al., 2002*). Altitude, ascending speed, and ascending mode, especially individual susceptibility, are the most important determining factors for the occurrence of HAPE (*Bartsch, 1999*).

A series of physiological responses across various body systems is involved in the process of HAPE. At high altitudes, hypoxia and inflammation can impair the function of the lungs, heart, and kidneys. Consequently, blood parameters and biochemical factors are reportedly as crucial to high-altitude acclimatization and disease management as other factors (*Bhandari & Cavalleri, 2019*; *Huang et al., 2017*; *Sanchez et al., 2022*). HAPE can trigger a severe inflammatory response or even death in later stages of its development if not appropriately and promptly treated (*El Alam et al., 2022*). However, current studies infrequently present a comprehensive analysis of the clinical characteristics and changes in blood biochemical indicators in patients with HAPE. Although chest X-rays and CT scans are the primary diagnostic methods for HAPE, many high-altitude regions have limited medical resources and lack specialized equipment. Additionally, HAPE can easily be confused with other lung injuries, leading to incorrect medication, a waste of medical resources, and complicating treatment. There are few statistical models for the integrated diagnosis of HAPE that incorporate physiological indicators, blood parameters, and biochemical factors. This study analyzed the laboratory indicator characteristics of HAPE patients, constructed a diagnostic model, and visualized it using a nomogram.

The diagnostic goal is to detect or identify HAPE in its early subclinical stage. Investigators continuously monitored individuals for evidence of HAPE based on specific clinical manifestations of the disease.

## MATERIALS AND METHODS

### Study design

This is a retrospective, cross-sectional study based on a medical center located in Lhasa, sited at 3,650 m on the Tibetan Plateau. The sample for the study was drawn from the population of patients attending to the Houbei Hospital in Lhasa from March 1, 2015 to May 31, 2022. The inclusion criteria of the subjects were as follows: (1) Han Chinese who are habitual low altitude dwellers; (2) had not previously suffered from high-altitude pulmonary edema; (3) without pulmonary disease (pulmonary embolism and pneumonia), cardiovascular disease (heart failure), serious mental disorder prior to entering high altitudes. Subjects younger than 18 years and with incomplete information in their profile were excluded from this study. We conducted non-experimental research using secondary data, anonymized patients, type and design with descriptive characteristics. The study procedures were approved by XIZANG Center for Disease

Control and Prevention ethics review committee conducted in accordance with the Helsinki Declaration (NO. 2023-001). We have obtained informed consent from the study participants, who have signed the consent form.

## Sample description

A total of 1,255 patients were enrolled, including 604 with HAPE (HAPE group) and 651 with non-HAPE (control group). We utilized the sample function in R software to randomly divide the dataset of 1,255 patients into two subsets: a training dataset and a validation dataset, maintaining a 7:3 ratio. To ensure reproducibility of the random split, we set the random seed to 666. This resulted in 864 patients in training data set and 391 patients in validation data set. The training data set was used to develop the prediction model and the nomogram, and the validation data set was utilized to assess the predictive performance of the model.

## HAPE diagnostics

The diagnosis of HAPE was based on both clinical reports and chest X-rays, as interpreted by medical professionals. HAPE is diagnosed in patients exhibiting symptoms including weakness, shortness of breath, cyanosis, dry cough upon exertion, pink frothy sputum, orthopnea, chest tightness, and fever. Meanwhile, characteristic X-ray features include interlobular pleural thickening, blurred lung textures, unilateral or bilateral lung lesions, reduced lung field translucency, and diffuse blurring (*Vock et al., 1989*).

## Physiological parameters measurement and collection

In accordance with non-probabilistic sampling methods, all data were extracted from patient files recorded between March 1, 2015, and May 31, 2022, and maintained in logbooks by authorized medical staff. Consistency between physical and digital records was ensured by two independent data entry personnel using double-entry methods. Demographic factors, physiological indices, hematological parameters, and biochemical indices were collected from all subjects on the first day after hospitalization, before any treatment was administered. Demographic variables included gender, age, and body mass index (BMI). Physiological parameters included heart rate (HR), breathing rate (BR), systolic blood pressure (SP), diastolic blood pressure (DP), and mean arterial pressure (MAP). Blood pressure was measured on the subject's left arm using an Omron digital BP monitor (Tokyo, Japan) or a manual sphygmomanometer when the automated monitor was unavailable.

Venous blood samples (3 ml) were collected using EDTA as an anticoagulant to determine hematological parameters. Hematological parameters included white blood cell (WBC) count, neutrophil percentage (NEUT%), lymphocyte percentage (LYMPH%), hemoglobin concentration (HGB), red blood cell (RBC) count, hematocrit (HCT), mean corpuscular volume (MCV), mean corpuscular hemoglobin (MCH), mean corpuscular hemoglobin concentration (MCHC), platelet count (PLT), mean platelet volume (MPV), procalcitonin (PCT), and platelet large cell ratio (PLCR). Hematological analysis was conducted using an Erma Hematology Analyzer (Tokyo, Japan). Serum sodium (Na),

potassium (K), chloride (Cl), and total calcium (tCa) levels were measured using a Hitachi 737 multichannel analyzer (Boehringer Mannheim Diagnostics, Indianapolis, IN, USA). To assess serum biochemistry, a venous blood sample (10 ml) was collected, and the precipitated serum was analyzed using a Mindray BS-220 blood biochemistry analyzer (China). Biochemical indices included alanine aminotransferase (ALT), aspartate aminotransferase (AST), total protein (TP), albumin (ALB), globulin (GLO), albumin/globulin ratio (A/G), total bilirubin (TBIL), direct bilirubin (DBIL), indirect bilirubin (IBIL), lactate dehydrogenase (LDH), and urea levels (UREA).

## Data preprocessing

We first reviewed data for range of values, completeness, and collinearity using R software (4.2.1; *R Core Team, 2022*). When the VIF is above 5, there might be a problem with multicollinearity. Individuals that had missing values, outliers and extremes in their records were removed from the dataset. Meanwhile, we converted two continuous variables, age and BMI, into categorical variables.

## Primary outcome and predictor selection

The primary outcome variable was the presence or absence of HAPE, coded as 1 and 0, respectively. Predictive variables included demographic factors, physiological indices, hematological parameters, and biochemical indices. To explore the relationship between the odds of HAPE and the factors under investigation, several generalized linear models (GLMs) with logit link functions were fitted to the data, assessing the relationship between HAPE (0 = absence; 1 = present) as the response variable (Y) and the explanatory variables (X). Variables for the final logistic regression model were meticulously selected using the Least Absolute Shrinkage and Selection Operator (LASSO) algorithm. LASSO, a widely recognized method for feature selection in scenarios involving high-dimensional data sets, has been extensively applied in statistical analyses and model building (*Tibshirani, 1997*). In this study, we implemented LASSO regression using the glmnet package, leveraging its robust capabilities for optimizing model accuracy and complexity. The Akaike Information Criterion (AIC) is a widely employed statistical model selection criterion that balances model complexity with goodness of fit. The fundamental principle of AIC is to minimize information loss between the model and the data, while preventing overfitting associated with overly complex models (*Burnham & Anderson, 2004*). For the final model, variables were carefully chosen to minimize the AIC value, ensuring an optimal trade-off between simplicity and accuracy. The training dataset was utilized for developing the prediction model in the logistic regression analysis. Subsequently, we constructed a nomogram to present the final model.

## Validation of the prediction model

Binary logistic regression models were employed to construct diagnostic models. We included gender, age, and BMI as covariates in the logistic models, but other varibles were included as continuous variables. The Hosmer-Lemeshow goodness-of-fit test and the validation sample were utilized to evaluate the calibration curve. The accuracy of the

model-predicted probabilities of diagnosing HAPE was evaluated using the Brier score. Internal validation of the predictive model was conducted using the bootstrap resampling method. Receiver operating characteristic (ROC) curves were evaluated for the discriminative power of the model. Decision curve analysis (DCA) was conducted to determine the clinical utility of the nomograms, quantifying the net benefits at various cutoff probabilities in the validation dataset (*Vickers & Elkin, 2006*).

## Statistical analysis

All statistical analyses were conducted using R software (version 4.2.1; *R Core Team, 2022*), SPSS (version 22.0), and STATA (version 17.0) for Windows (StataCorp, College Station, TX, USA). The Mann-Whitney U test was utilized for comparing independent samples, and the Wilcoxon signed-rank test was employed for pairwise comparisons. Categorical data are presented as percentages, while continuous data are shown as medians (25th to 75th percentiles). We report odds ratios (ORs) and 95% confidence intervals (95% CIs) assessed from multiple logistic regression models adjusting for age, gender, and BMI. The reported levels of statistical significance are all two-tailed, with a significance threshold set at $P < 0.05$.

# RESULTS

## Clinical characteristics

From March 2015 to May 2022, 1,255 subjects were screened for eligibility (Fig. 1). The hospital-based sample consisted of 842 males and 413 females, aged 18 to 76 years. Of these, 604 subjects were admitted to the emergency room of Houbei Hospital a few days after arriving at 3,500 m above sea level, presenting significant symptoms of altitude sickness, and were later hospitalized with a diagnosis of HAPE. The control group comprised 651 subjects who were hospitalized at Houbei Hospital during the same period and met the inclusion criteria. Within the data framework, all participants self-identified as Han Chinese, having lived on the plains for an extended period. No indigenous Tibetans were included in the analyzed population. The confidence level for all sampled data was set at 95%. Except for age, chloride (Cl), globulin (GLO), total bilirubin (TBIL), and direct bilirubin (DBIL), the remaining indices differed between the HAPE and control groups. Subsequent analyses were conducted after excluding indicators that showed no differences. Of the 604 HAPE patients, 76.37% were men, 1.99% were aged over 65 years, and 47.85% were overweight (Table S1).

## Feature selection

Data from 864 patients were utilized for the development of the nomogram prediction model, while data from 391 patients were used to evaluate its performance. Table S2 displays the characteristics of the patients in the training and validation datasets. Typically, the sample should include a minimum of 10 outcome events per variable (EPVs) to provide robust estimates (*Peduzzi et al., 1995*; *Peduzzi et al., 1996*). Our sample and the number of events exceeded those recommended by the EPV approach for determining sample size in the training set compared to the validation set. There were no significant

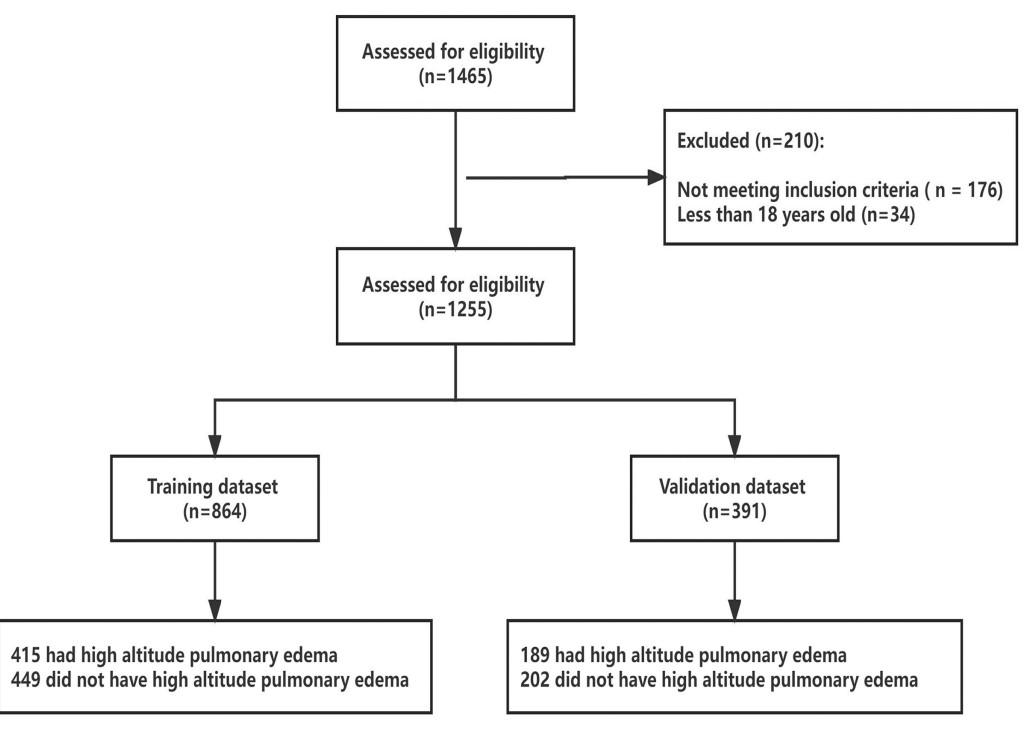

**Figure 1 Enrolled subjects and outcome of HAPE in training and validation datasets.**

differences between the two data sets for any variable. In the training and validation datasets, the proportion of patients with HAPE was 48.03% and 48.34%, respectively.

## Development of an individualized diagnostic model

From a total of 34 features, 11 potential predictors were identified from 864 patients (Figs. 2A and 2B), characterized by having non-zero coefficients in a LASSO logistic regression model. The model, incorporating the independent predictors mentioned above, was developed (Table S3) and is presented as a nomogram (Fig. 3).

## Validation of the HAPE nomogram

**Discrimination:** Figure 4 illustrates the area under the ROC curve (AUC). The nomogram demonstrated good discrimination, with AUC values of 0.787 (95% CI [0.757–0.817]) for the training dataset and 0.833 (95% CI [0.793–0.874]) for the validation dataset.

   **Calibration:** The calibration curve, displayed in Figs. 5A and 5B, was evaluated using the U-test, yielding $P$-values of 0.793 for the training dataset and 0.629 for the validation dataset. The non-significance of the Hosmer-Lemeshow test indicates a high level of agreement. The results of the calibration curve demonstrate that the predicted probabilities from the model align closely with the actual observed outcomes, indicating strong consistency and reliability of the model's predictions.The accuracy of the predictions was assessed using the Brier Score (BS), defined as the mean squared deviation between the predicted probabilities and the actual outcomes. The Brier Score was 18.7% for the training set and 18.0% for the validation set, as shown in Figs. 5A and 5B. Meanwhile, a BS of 24.3%

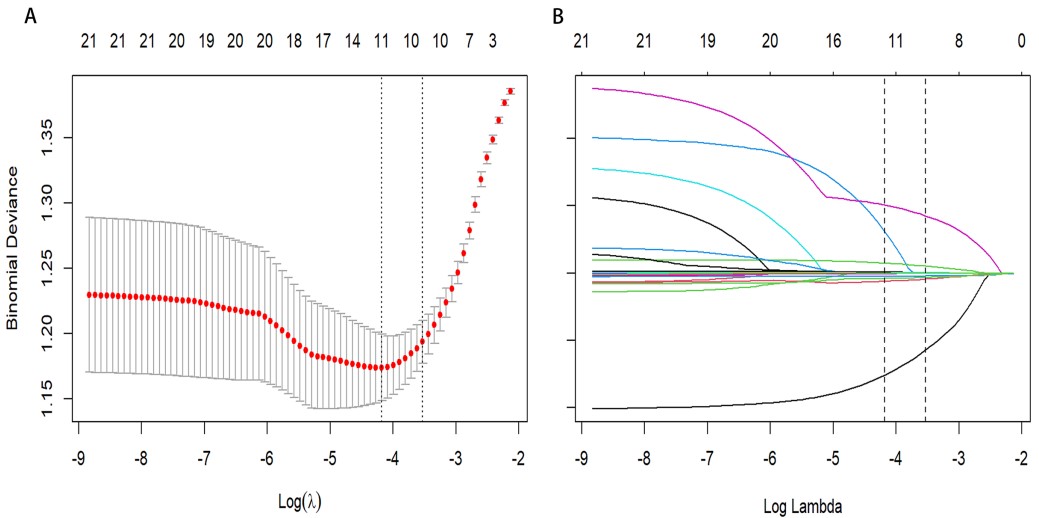

**Figure 2 Predictor selection using the least absolute shrinkage and selection operator (LASSO) binary logistic regression model.** (A) Identifying the optimal penalising coefficient lambda ($\lambda$) in the LASSO model using tenfold cross-validation and the minimum criterion. (B) LASSO coefficient profiles of 11 variables. The value selected by 10-fold cross-validation was plotted as a vertical line. As $\lambda$ decreased, the degree of model compression increased and the model's ability to select important variables increased.

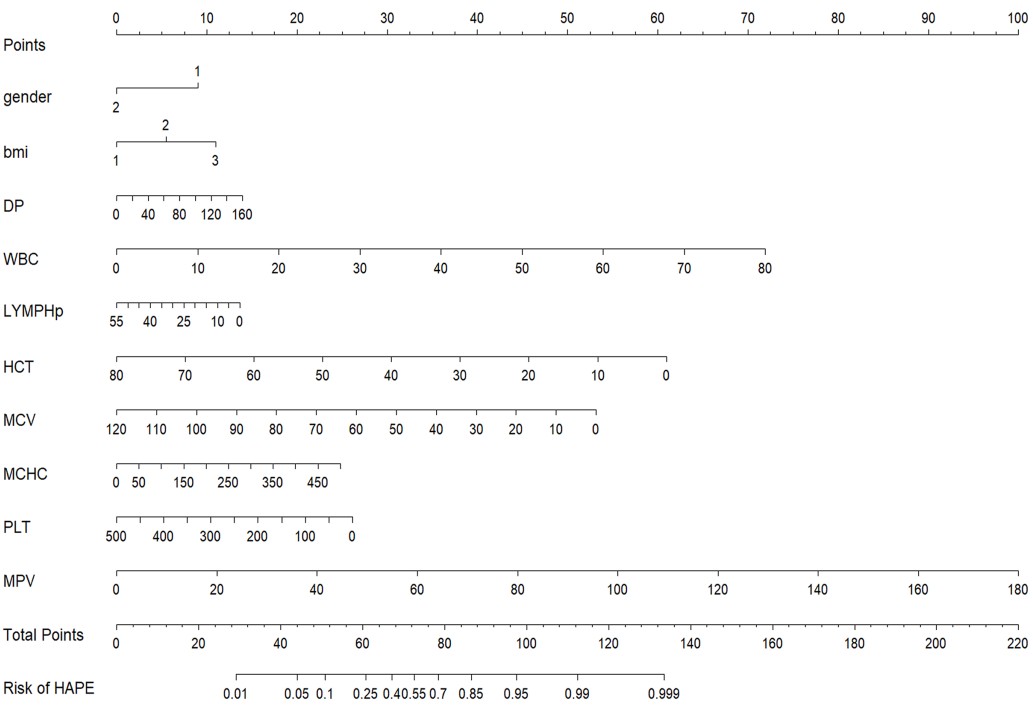

**Figure 3 Nomogram to estimate the probability of HAPE.** Predictor scores associated with each patient variable are obtained and summarised on the upper scale. The percentage probability of HAPE is calculated from the sum of the scores projected onto the lower scale.

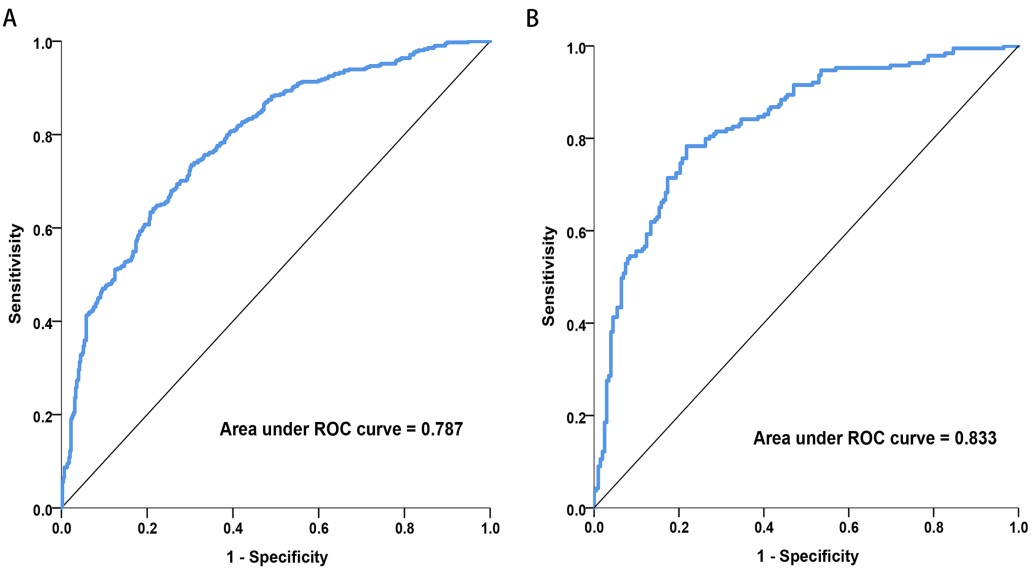

**Figure 4 Receiver operating characteristic (ROC) curves of the nomograms in training and validation dataset.** (A) Training dataset. (B) Validation dataset. The nomogram had good discriminative power with area under ROC curve (95% confidence interval) of 0.787 (95% CI [0.757–0.817]) and 0.833 (95% CI [0.793–0.874]) in the training and validation dataset, respectively.

was calculated through bootstrap resampling of 100 replicates (Fig. 5C). Confusion matrices, compiled for both datasets, indicated accuracies of 70.95% for the training set and 74.17% for the validation set (Table S4). The nomogram-based predictions showed good agreement with the observations from the calibration curve regarding the probability of HAPE.

## Clinical use

The clinical utility and net benefit of the nomogram are demonstrated by the decision curve analysis shown in Fig. 5D. The decision curve indicated that when the threshold probability exceeded 10%, using the nomogram to predict HAPE offered more benefit compared to treating all patients as either HAPE or non-HAPE, particularly for threshold probabilities ranging from 10% to 95%.

## DISCUSSION

HAPE typically occurs and progresses rapidly. If not treated promptly and effectively, HAPE can deteriorate into multiple organ dysfunction syndrome, potentially complicating treatment or resulting in death (Choudhary et al., 2023). Prompt diagnosis is essential to prevent the condition from worsening. An accurate diagnostic model enables healthcare professionals to detect HAPE in its early stages or before symptoms emerge, allowing for earlier intervention, thus averting severe complications and lowering mortality rates. Diagnosis *via* laboratory markers is generally simpler and more convenient than imaging studies or other invasive tests. If the model can be based on standard blood and biochemical tests, it will significantly simplify and lower the cost of diagnosis. In this study,

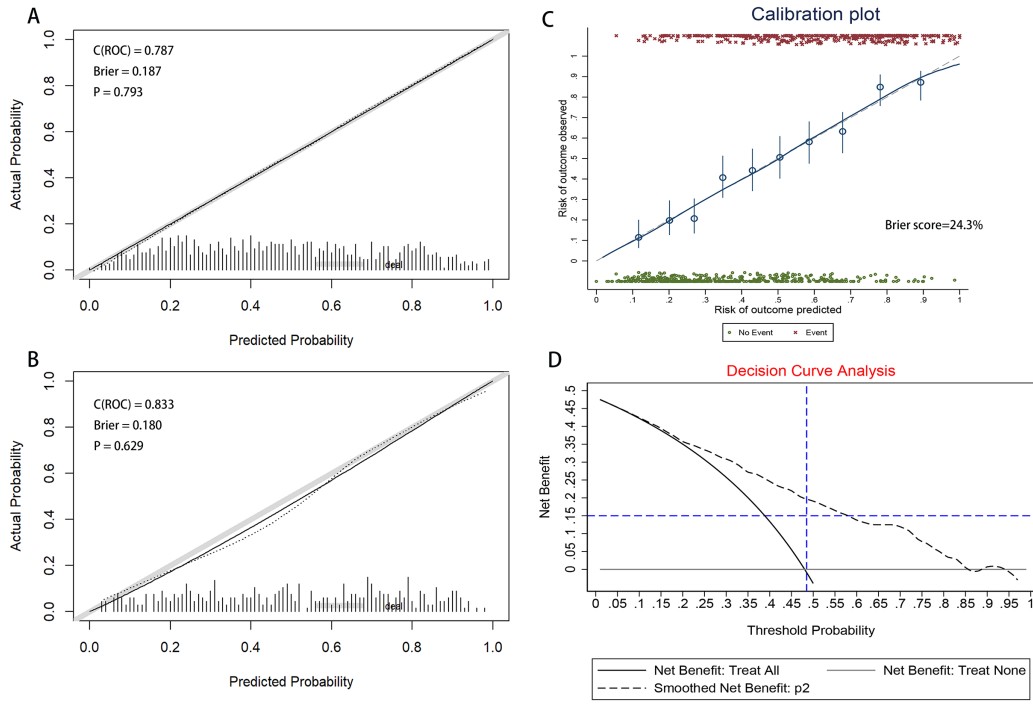

**Figure 5 The calibration curve of nomogram for predicting HAPE in the training and validation dataset.** (A) Training dataset. (B) Validation dataset. Calibration focuses on the accuracy of the model in predicting absolute risk, *i.e.*, the consistency between what the model predicts will happen and what actually happens. The y-axis is the actual rate of HAPE. The x-axis is the predicted probability of HAPE. The y-axis represents the actual rate of HAPE. The x-axis is the predicted probability of HAPE. In a well-calibrated nomogram, the scatter points should be aligned along a diagonal line at a 45 degree angle. $P > 0.05$ means no significant difference and the model is well calibrated. (C) Calibration plot of training dataset that the BS was calculated by bootstrap resampling of 100 replicates. (D) Decision curve analysis for the nomogram. The y-axis measures the net benefit. The black dotted line represents the nomogram. The solid black line represents the assumption that all patients have HAPE. The grey line represents the assumption that no patients have HAPE. The net benefit was calculated by subtracting the proportion of all patients with a false-positive result from the proportion with a true-positive result, weighted by the relative harm of not receiving treatment compared with the negative consequences of receiving unnecessary treatment (*Vickers & Elkin, 2006*). The decision curves showed that the use of the nomogram in the current study for predicting HAPE gave more benefit than either the "treat all patients" or the "treat none" regimen when the patient or physician threshold probability exceeded 10%.

we analyzed changes in clinical characteristics and biochemical markers among Han Chinese who traveled to high-altitude destinations to determine if these significant changes are associated with HAPE. Our study further explored the diagnostic model for HAPE, utilizing demographic factors, physiological indices, hematological parameters, and biochemical indices. To our knowledge, this is a unique diagnostic model that reports the occurrence of HAPE in Han Chinese following rapid ascent to the Tibetan Plateau.

The results indicated that the most prevalent clinical and laboratory changes in HAPE patients, resulting from high-altitude exposure, were related to blood cells, immune cells, liver, and heart functions. Utilizing these clinical and laboratory changes, a diagnostic nomogram for individualized prediction of HAPE has been developed and validated. For the development of the HAPE signature, 34 candidate features were reduced to

11 potential predictors by assessing the associations between predictors and outcomes and shrinking the regression coefficients *via* the LASSO method. The nomogram incorporates eleven items, including gender, BMI, DP, MAP, WBC, LYMPH%, HCT, MCV, MCHC, PLT, and MPV. This predictive model demonstrated adequate discrimination in the training cohort (AUC: 0.787), with a surprising improvement in the validation cohort (AUC: 0.833). The chosen model, which boasts a low Brier score, was computed using the bootstrap resampling method and validated in both the training and validation sets. Given that lower Brier scores indicate higher model accuracy, the model with the lowest Brier score was ultimately selected as the final model (*Seif et al., 2014*).

Interestingly, only blood parameters, along with gender, age, BMI, and blood pressure, were needed to make predictions about HAPE. Health systems in high-altitude areas are frequently characterized by inequitable distribution, limited medical resources, transport difficulties, and the isolation of small regions (*Li et al., 2012*). As detailed in the results, the nomogram developed in our study predicts the likelihood of an individual developing HAPE by solely combining blood parameters and demographic factors. A nomogram integrates multiple predictive indicators, representing them on a single plane through scaled line segments to illustrate the interrelationships among variables in the predictive model. This visualization, typically arranged as column charts, simplifies complex regression equations, making the outcomes of prediction models more intuitive and accessible. Such design enhances understandability, allowing even non-professionals to easily interpret and utilize the information. Users can swiftly estimate the predicted outcomes for high-altitude pulmonary edema (HAPE) under specific conditions by simply aligning values and reading off the chart. Employing only routine blood tests for diagnosis, without excessive laboratory testing, reduces patient costs. This approach not only improves clinical decision-making for clinicians and patients but also results in a greater net benefit. In line with the current trend towards personalized medicine, this user-friendly scoring system enables both clinicians and patients to make individualized predictions of HAPE risk (*Balachandran et al., 2015*). Our diagnostic model enables early preventive interventions for patients with HAPE and, conversely, prevents the overtreatment or biased treatment of those without HAPE. Numerous retrospective observational analyses have described the complex interplay of various erythrocyte indices in acclimatization and adaptation to high-altitude hypoxia. Exposure to hypoxia at high altitudes is known to stimulate increased bone marrow production of red blood cells (erythropoiesis) and the formation of new blood vessels (angiogenesis) (*Mallet et al., 2023*). Research indicates that in the acute phase, blood cell counts such as RBC, HGB, and HCT significantly increase and continue to do so in the chronic phase, while platelet counts are significantly reduced, benefiting blood flow (*Azad et al., 2017*; *Srihirun et al., 2012*). In the chronic phase, the volume of red blood cells and MCH continue to increase, thereby enhancing the blood's oxygen-carrying capacity (*Sanchez et al., 2022*). Additionally, neutrophils and platelets are simultaneously activated to regulate the inflammatory response during acute inflammation (*Daseke et al., 2019*). Although mechanistically HAPE is not triggered by inflammation, its occurrence can lead to inflammatory complications within the body (*Idzko et al., 2014*). In our study, both white blood cell count and total neutrophil percentage are critical
indicators for predicting the inflammatory response that accompanies most cases of HAPE. Routine blood tests are instrumental not only in diagnosing HAPE but also in monitoring changes in a patient's condition during clinical treatment. Regular assessments enable physicians to track critical parameters such as oxygen levels, red blood cell counts, inflammatory markers, and other indicators included in the diagnostic model. Variations in these metrics can reveal the patient's response to treatment and provide prognostic insights. Timely adjustments to treatment strategies—such as modifying oxygen supplementation, pharmacotherapy, and supportive care—can significantly enhance patient outcomes and reduce the risk of complications.

We also found that HAPE occur more frequently in males compared with female. Some studies demonstrated that females are less susceptible to high-altitude sickness and are better adapted to high altitude than males (*Hackett et al., 1998*). The reason for this may be that estrogen dilates the renal microvascular system, increases the hematocrit in blood from small arteries, capillaries and veins, alters oxygen delivery per unit of erythrocyte mass and provides a mechanism for altering erythrocyte mass without compensatory erythropoiesis (*Murphy, 2014*). Considering potential uncorrected gender differences in logistic regression models, we conducted separate analyses for each gender. The results demonstrated that the area under the curve (AUC) for the three datasets—the total dataset, male dataset, and female dataset—were 0.797 (95% CI [0.773–0.822]), 0.796 (95% CI [0.766–0.827]), and 0.817 (95% CI [0.775–0.859]) respectively. These findings indicate that age and gender heterogeneity have minimal impact on the diagnostic outcomes, as detailed in Table S5. In addition, overweight or obese people are more prone to HAPE than slimmer people in this study. A few of studies have focused on the relationship between BMI and high-altitude acclimatization (*Coello et al., 2000*; *Shen et al., 2017*), but there is scarce study focus on the relationship between BMI and HAPE. It was indicated that subjects with higher BMI responses reluctant to hypoxia. Acutely, blood pressure, respiratory function and oxygen saturation increased less in people with higher BMI than in those with lower BMI (*Peng et al., 2013*). The reason for this phenomenon may be that in subjects with a higher BMI, there is little room for blood pressure or respiratory function increase in the acute phase because they were already higher at baseline. Also, blood pressure and red blood cell volume increased less in people with a higher BMI than in people with a lower BMI. Human blood pressure (BP) homeostasis is challenged by hypoxia at high altitudes. The effects of altitude exposure on patients with hypertension have also been the subject of specific studies. Acute exposure to hypobaric hypoxia at high altitudes induces an increase in blood pressure in humans with high blood pressure, both at rest and during exercise (*Caravita et al., 2018*). In our study, DP was included in the model, suggesting that blood pressure is also an indicator of great interest in studying HAPE in Han Chinese. Exposure to high altitude is associated with significant changes in cardiovascular function, and this is probably the reason.

This study retrospectively analyzed previous case data to explore the disease characteristics and predictive factors of high-altitude pulmonary edema. Although this research method is widely used in medical research, there are also some inherent limitations. Firstly, data relies on existing medical records and faces the problem of

incomplete data. Some important information may not be recorded or recorded incompletely. Secondly, this study is based on patients in a medical center, and the results may affect the promotion and application of the research findings in other populations.

## CONCLUSION

In conclusion, we have presented clinical characteristics and a nomogram based on hematological parameters and demographic variables, which can be conveniently used for individualized diagnosis of HAPE. In high-altitude areas with limited emergency environments or resources, diagnostic models based on laboratory indicators can provide fast and reliable diagnostic support for medical staff, helping them make better treatment decisions. Moreover, the symptoms of high-altitude pulmonary edema may be similar to other high-altitude diseases or respiratory system diseases, which can easily lead to misdiagnosis. An accurate diagnostic model can reduce the risk of misdiagnosis and improve the accuracy of clinical diagnosis. By utilizing machine learning technology to process and analyze a large amount of patient laboratory data, and establishing and optimizing such diagnostic models, we can better understand the pathogenesis and risk factors of HAPE, providing data support for future research and drug development.

## ABBREVIATIONS LIST

| | |
|---|---|
| **A/G** | albumin/globulin ratio |
| **AIC** | Akaike's information criterion |
| **ALB** | albumin |
| **ALT** | alanine aminotransferase |
| **AST** | aspartate aminotransferase |
| **AUC** | the area under the ROC curve |
| **BMI** | body mass index |
| **BR** | breathing rate |
| **BS** | Brier Score |
| **Cl** | chloride |
| **DBIL** | direct bilirubin |
| **DCA** | Decision curve analysis |
| **DP** | diastolic blood pressure |
| **EPVs** | events per variables |
| **GLMs** | generalized linear models |
| **GLO** | Globulin |
| **HACE** | high-altitude cerebral edema |
| **HAPE** | High-altitude pulmonary edema |
| **HCT** | hematocrit |
| **HGB** | hemoglobin |
| **HR** | heart rate |
| **IBIL** | Indirect bilirubin |
| **K** | potassium |

| | |
|---|---|
| **LASSO** | least absolute shrinkage and selection operator |
| **LDH** | lactate dehydrogenase |
| **LYMPH%** | lymphocyte percentage |
| **MAP** | mean arterial pressure |
| **MCH** | mean corpuscular hemoglobin |
| **MCHC** | mean corpuscular hemoglobin concentration |
| **MCV** | mean corpuscular volume |
| **MPV** | mean platelet volume |
| **Na** | sodium |
| **NEUT%** | neutrophil percentage |
| **PCT** | hematocrit |
| **PLCR** | platelet-larger cell ratio |
| **PLT** | platelet |
| **RBC** | red blood cells |
| **ROC** | Receiver operating characteristic curves |
| **SP** | systolic blood pressure |
| **TBIL** | total bilirubin |
| **tCa** | total calcium |
| **TP** | total protein |
| **UREA** | urea levels |
| **WBC** | white blood cell |

# ACKNOWLEDGEMENTS

We thank all the subjects of this study and the staff of the Center for Disease Control and Prevention of the XiZang and the Houbei Hospital in Lhasaon the Qinghai-Tibetan Plateau.

## Funding

The authors received no funding for this work.

## Competing Interests

The authors declare that they have no competing interests.

## Author Contributions

- Qiong Li conceived and designed the experiments, performed the experiments, prepared figures and/or tables, authored or reviewed drafts of the article, and approved the final draft.
- Zhichao Xu performed the experiments, analyzed the data, prepared figures and/or tables, and approved the final draft.

- Qianhui Gong performed the experiments, analyzed the data, prepared figures and/or tables, and approved the final draft.
- Xiaobing Shen conceived and designed the experiments, authored or reviewed drafts of the article, and approved the final draft.

### Human Ethics

The following information was supplied relating to ethical approvals (*i.e.*, approving body and any reference numbers):

The study procedures were approved by the Xizang Center for Disease Control and Prevention ethics review committee conducted in accordance with the Helsinki Declaration of 1975 as revised in 2000 (NO. 2023-001).

### Data Availability

The raw measurements are available in the tables.

### Supplemental Information

Supplemental information for this article can be found online at http://dx.doi.org/10.7717/peerj.18084#supplemental-information.

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
