# Peer review of "Clinical characteristics and a diagnostic model for high-altitude pulmonary edema in habitual low altitude dwellers"

_PeerJ, doi:10.7717/peerj.18084_

## Round 0.1 · original submission · Major Revisions

The two expert reviewers have noted the quality of your article and its relevance in identifying HAPE in settings with limited medical resources. They provided valuable insights to improve its methodological and conceptual foundations.

Please also improve the quality and legibility of figures (especially since the open access format of peerj allows creativity in this domain). The tables are exceedingly long: I question whether this should be provided as a downloadable table along with the study dataset made available on a data repository with permanent DOI.

I would also add as a minor comment that the characteristics of the population (lowland dwellers) should be explicited more plainly (in the title, aims of the study, methods, L81 : not sure of the expression "survived in the lowlands", probably more something like "habitual low altitude dwellers" or at least that they are not habitual mid or high altitude dwellers).

I recommend for major revisions since the scope of the required amendments are significant and will have a major impact on the quality of the article.

Reviewer 1 ·

Basic reporting

1. The current approach achieves a maximum diagnostic accuracy of approximately 80% for HAPE. Discuss potential avenues for further analysis, such as stratifying samples by age groups and conducting separate analyses for males and females, to evaluate the impact of age and gender heterogeneity on diagnostic outcomes. However, given the observed 80% accuracy, elucidate the uncertainties regarding the clinical utility of the proposed approach.

2. Discuss the relevance of the study findings to populations other than Han Chinese. Alternatively, consider revising the title to specify "Han Chinese descent" to align with the demographic focus of the study and explicitly clarify the target population.

3. The analysis provided does not assess the severity of HAPE or the timing of disease onset, which limits the early diagnostic capabilities of the proposed approach.

4. Expand on the implications of the study findings for clinical practice in high-altitude regions. Emphasize how the nomogram’s simplicity and reliance on routine blood tests can potentially enhance diagnostic accuracy and improve patient outcomes.

Experimental design

1. Provide information on the timing of HAPE onset and sample collection after arrival in high-altitude (HA) areas.

2. Enhance clarity in interpreting the calibration curves, and their implications for model reliability in both training and validation datasets.

3. Provide a detailed description of how predictors were selected using LASSO regression. Justify the use of Akaike’s Information Criterion (AIC) as the stopping rule and explain why these specific predictors were considered relevant.

4. Clarify the rationale behind using the split-sample method to divide the dataset into training and validation sets.

Validity of the findings

Although the study highlights improvements in accurate treatment processes, it lacks an analysis demonstrating that the observed outcomes or provided nomogram are specific to HAPE and not applicable to other high-altitude diseases or respiratory system diseases, potentially leading to misdiagnosis. Therefore, the current study may have limited contribution towards minimizing false positive HAPE diagnoses in clinical practice.

Additional comments

Quality and visibility of figures can be improved.

·

Basic reporting

This paper is well written, and follows a logic behind research presented.
Grammar check is strongly recommended.

Experimental design

Methodology described seems to be sound and follows standard practices in statistical modeling and machine learning. However, there are a few areas that could potentially be improved or clarified:

Data Preprocessing: The excerpt does not mention any data preprocessing steps. It’s important to handle missing data, outliers, and ensure that the data is appropriately scaled or transformed.

Interpretability: While LASSO is a good choice for feature selection, it might make the model less interpretable, especially if many features are selected. If interpretability is a concern, other methods or additional steps might be needed to better understand the model.

Multicollinearity: In the presence of highly correlated predictors, LASSO might select one at random, which could lead to unstable models. It would be good to check for multicollinearity and possibly address it before model fitting.

The median is the middle value that separates the higher half from the lower half of a data sample, and it is less affected by outliers and skewed data than the mean. The 25th percentile (also known as the first quartile) is the value below which 25% of the observations fall, and the 75th percentile (or third quartile) is the value below which 75% of the observations fall. You must clarify this "as medians (25th to 75th percentiles)."

Validity of the findings

The study has high clinical relevance, as it aims to develop a diagnostic model for HAPE, a potentially life-threatening condition. The model is based on demographic factors, physiological indices, hematological parameters, and biochemical indices, which are generally easier and more cost-effective to measure than imaging studies or other invasive tests.

The validity of the findings in this study seems to be well-supported by the data and the statistical analyses performed. Here are some key points to consider:

Statistical Analysis: The statistical analyses appear to be apppropiated. The use of the Area Under the ROC Curve and Brier Score for assessing the performance of the model is appropriate. The AUC values indicate good discrimination ability of the model, and the Brier scores, which were low, indicate high accuracy of the predictions. However, I recommend to perform full confusion matrix analysis.

Additional comments

No additional comments.

---

## Round 0.2 · accepted · Accept

Thank you for adressing the expert reviewers' recommendations, and providing an improved article that is now acceptable for publication.

Reviewer 1 ·

Basic reporting

Response to queries is satisfactory.

Experimental design

Response to queries is satisfactory.

Validity of the findings

Response to queries is satisfactory.

Additional comments

No further query.

·

Basic reporting

Comments and revisions on areas where the article was improved meets journal standards.

Experimental design

Comments and revisions on areas where the article was improved meets journal standards.

Validity of the findings

Comments and revisions on areas where the article was improved meets journal standards.

Additional comments

NA